# OpenReview forum: "Follow My Instruction and Spill the Beans: Scalable Data Extraction from Retrieval-Augmented Generation Systems"
_ICLR.cc/2025/Conference — ICLR 2025 Poster_

### Official Review · Reviewer_H6ab · 2024-11-01

**Soundness:** 4
**Presentation:** 3
**Contribution:** 3
**Rating:** 8
**Confidence:** 4

**Summary:**

This paper explores the vulnerability of Retrieval-Augmented Generation (RAG) systems to data extraction attacks. The authors demonstrate that adversaries can exploit the instruction-following capabilities of language models to extract sensitive data using prompt injection techniques.  The paper also examines the impact of RAG setup on data extractability and proposes mitigation strategies, such as position bias elimination. The contribution of this work provided suggestion from both rag and model view.

**Strengths:**

This is a very meaningful research. It is important to understand how LLM and RAG can be attacked. I also saw some methods to cheat the benchmark and maliciously increase the score. The exploration of the RAG system is very novel. They also explored different attacker policy and methods such as black-box access and instruction injection.

**Weaknesses:**

The proposed method in the paper is simple and seems to be effective, but it seems it's more about prompt engineering. Without improving the causal LLM, is this really a necessary approach to solving the problem?

**Questions:**

Q: Recent work has explored methods for test-time reasoning. Perhaps considering allowing the model to select/punish tokens non-greedily during the inference phase can also alleviate the problem of information leakage?

---

> ### Author Response · Authors · 2024-11-20
> **Response to Reviewer H6ab**
>
> We thank the reviewer for their feedback and questions.
>
>
> > The proposed method in the paper is simple and seems to be effective, but it seems it's more about prompt engineering. Without improving the causal LLM, is this really a necessary approach to solving the problem?
>
> We appreciate this observation. We agree that improvements to the LLM itself, such as through architectural changes or enhanced training objectives, represent a promising long-term solution. These advancements could complement prompt-level defenses and potentially mitigate the vulnerabilities at their root. While our work leverages prompt engineering as a critical component, it is designed to highlight the vulnerabilities in widely-used RAG systems, particularly those stemming from instruction-tuned LLMs. The simplicity of our approach ensures generalizability across various LLMs and RAG setups without requiring changes to the underlying model. This practicality is a strength, as it allows for immediate testing and mitigation strategies on existing systems, and thus is necessary. Moreover, our usage of the positional bias elimination method (PINE) could count as as method of improving the causal LLM.
>
> > Recent work has explored methods for test-time reasoning. Perhaps considering allowing the model to select/punish tokens non-greedily during the inference phase can also alleviate the problem of information leakage?
>
> This is an insightful suggestion. Non-greedy inference strategies could indeed mitigate leakage by reducing the model's propensity to directly copy sensitive retrieved content. In our study, however, we focused on identifying vulnerabilities under the most typical inference setups that are widely adopted. We do recognize the potential of inference-phase modifications to enhance robustness against leakage and will include such directions to our future work.

---

> > ### Comment · Reviewer_H6ab · 2024-11-25
> >
> > I would like to thank the authors for addressing most of my concerns. Hope they would involve the change in their revised version paper. Due to these factors I have chosen to maintain my score.

---

### Official Review · Reviewer_VHqF · 2024-11-02

**Soundness:** 3
**Presentation:** 3
**Contribution:** 3
**Rating:** 6
**Confidence:** 2

**Summary:**

This paper examines how Retrieval-Augmented Generation (RAG) systems can be exploited through prompt injection attacks to extract data from their datastores. The authors demonstrate that instruction-tuned language models are particularly vulnerable to these attacks, with larger models being more susceptible. They test their approach on both open-source models and production systems (GPTs), achieving concerning success rates.

Detailed Comments:

The experimental methodology is sound, if straightforward. The authors test their attack across different model sizes, architectures, and configurations. The ablation studies examining factors like context position and chunk size are well-executed and informative.

The GPT results are interesting, though I suspect OpenAI will patch this particular attack vector about 5 minutes after this paper is published (if they haven't already). This highlights a broader issue with the paper - it feels more like a bug report than a fundamental research contribution.

Section 3's analysis of how different factors affect extraction success is probably the strongest technical contribution, but even this feels more like engineering characterization than novel research insights.

The writing has this breathless "we discovered a vulnerability!" tone that doesn't quite match the technical depth of what's actually being presented. Yes, if you tell a helpfulness-optimized AI to be helpful by sharing information with you, it tends to do that. This is about as surprising as discovering that scissors can be used to cut things they weren't supposed to cut.

That said, I appreciate the thorough empirical work and clear presentation. The paper does provide a useful characterization of how various factors influence RAG systems' susceptibility to data extraction. The mitigation strategies section, while basic, provides a starting point for thinking about solutions. I don't like this paper, not one bit - but I can't bring myself to say "marginally below acceptance threshold" on good work, even if I don't like it. I don't come from this security research world.

While well-executed, this paper falls just above of the bar for acceptance. The core observation is obvious, and the technical contributions aren't quite deep enough for me to justify anything except a marginal accept. This would make an excellent blog post or technical report, but needs more novel technical insights or fundamental theoretical contributions for me to be comfortable with this publication.

The authors should consider:
- Exploring fundamental architectural solutions
- Providing theoretical analysis of the trade-offs between retrieval utility and data protection
- Examining this in the context of broader information security frameworks

In its current form, this feels more like a well-documented proof-of-concept than a research paper advancing the field's understanding of language model security.

**Strengths:**

+ Comprehensive empirical evaluation across multiple models and scales
+ Clear ablation studies examining various factors affecting extraction success
+ Practical demonstration on production systems (GPTs)
+ Thoughtful analysis of mitigation strategies
+ Good technical writing and structure

**Weaknesses:**

The core attack is, frankly, obvious to anyone who has worked with RAG systems - you just tell the model to repeat its context. This feels more like a blog post documenting "look what I found" than novel research. The authors basically discovered that if you put private data in front of an instruction-following AI and tell it to repeat that data, it... repeats the data. Color me shocked.

While the paper presents this as a novel security vulnerability, this is really just documenting an obvious limitation of current RAG architectures. It's like publishing a paper saying "if you give someone your house key, they can open your door." The fact that instruction-tuned models follow instructions shouldn't be presented as a surprising security flaw.

The proposed mitigation strategies feel underdeveloped. Position-bias elimination and safety-aware prompts are reasonable starting points, but the paper doesn't deeply engage with fundamental architectural changes that might be needed. It's like putting a "please don't enter" sign on an unlocked door and calling it security. I'm not compelled especially by the results - but I have no evidence against them.

**Questions:**

1. How do you envision RAG systems fundamentally preventing this kind of attack while maintaining their core functionality?
2. Have you considered more sophisticated architectural approaches to information compartmentalization?
3. How do you respond to the criticism that this is simply documenting an obvious limitation rather than a novel security vulnerability?

---

> ### Author Response · Authors · 2024-11-20
> **Response to Reviewer VHqF (Part 1)**
>
> Thank you for your detailed and thoughtful feedback. We value your insights and appreciate the opportunity to clarify our contributions. Below, we address each point raised in your review.
>
> > The core attack feels obvious. How do you respond to the criticism that this simply documents an obvious limitation?
>
> While we acknowledge that the prompt-injection attack on RAG systems described seem intuitive, we respectfully argue that our work extends far beyond merely documenting a known issue or posting a blog in several key ways:
>
> - Formal Problem Definition: We formally define prompt-injected data extraction, framing it as a generalizable threat rather than an isolated instance.
> - Systematic Study: Our study is the first to systematically explore and quantify this vulnerability across a wide range of model scales and RAG configurations. This approach goes beyond anecdotal observations or blog posts by providing controlled experiments, comprehensive ablations, and formal problem definitions. For instance, we examine how variables like model size, context position, and chunking strategies directly influence extractability, offering new insights into these systems' susceptibility.
> - New Insights: We especially highlight the interplay between position bias and vulnerability to prompt injection, identifying it as a critical factor influencing data leakage. This observation is novel and contributes meaningfully to the broader understanding of RAG vulnerabilities. Moreover, based on this finding, we propose the potential mitigation method which is proved to be effective.
> - Practical Mitigations: We provide concrete, simple, and reproducible mitigation strategies and evaluate their efficacy, setting the stage for further work in more robust defense methods.
> - Broader Implications: By testing production systems like GPTs, we demonstrate the practical significance of the vulnerability brought by strong instruction following abilities in real-world applications. This is not merely a hypothetical concern but a pressing issue for deployments where RAG systems are integrated into applications dealing with sensitive, confidential, or proprietary data., especially as larger models become more prevalent.
>
> Similar to many other well-acknowledged empirical works in ML security [1,2,3,4,5], our study aims to emphasize the importance of a thorough exploration of a security issue, even though it is straightforward, and demonstrates baseline mitigations as practical starting points.
>
> [1] Carlini, N., Tramer, F., Wallace, E., Jagielski, M., Herbert-Voss, A., Lee, K., ... & Raffel, C. (2021). Extracting training data from large language models. In 30th USENIX Security Symposium (USENIX Security 21) (pp. 2633-2650).
>
> [2] Carlini, N., Hayes, J., Nasr, M., Jagielski, M., Sehwag, V., Tramer, F., ... & Wallace, E. (2023). Extracting training data from diffusion models. In 32nd USENIX Security Symposium (USENIX Security 23) (pp. 5253-5270).
>
> [3] Nasr, M., Carlini, N., Hayase, J., Jagielski, M., Cooper, A. F., Ippolito, D., ... & Lee, K. (2023). Scalable extraction of training data from (production) language models. arXiv preprint arXiv:2311.17035.
>
> [4] Huang, Y., Gupta, S., Zhong, Z., Li, K., & Chen, D. (2023). Privacy implications of retrieval-based language models. arXiv preprint arXiv:2305.14888.
>
> [5] Lukas, N., Salem, A., Sim, R., Tople, S., Wutschitz, L., & Zanella-Béguelin, S. (2023, May). Analyzing leakage of personally identifiable information in language models. In 2023 IEEE Symposium on Security and Privacy (SP) (pp. 346-363). IEEE.

---

> ### Author Response · Authors · 2024-11-20
> **Response to Reviewer VHqF (Part 2)**
>
> > Feels like a bug report rather than novel research
>
> We appreciate your perspective, but hope to emphasize that our work goes beyond being a mere "bug report". It demonstrates how strong instruction-following capabilities can be strategically exploited to compromise datastore confidentiality. Our proposed attack on GPTs is just one simple example of how to leverage the capability of LLM to attack itself. Even though OpenAI fixes the bug right away, there might still be many other side channels stemming from the interactions between the user and the backbone instruction-tuned LLM, which can be used to attack the system.
> The vulnerability we highlight is not just a simple flaw but a reflection of a deeper trade-off: optimizing systems for instruction-following inadvertently creates pathways for exploitation. While the attack is straightforward, it serves as a critical reminder for RAG deployers to address these risks proactively.
> Again, we assert that this work is not just a discovery but a systematic study that formalizes the problem, evaluates its scope and impact, and explores potential mitigations.
>
> > Proposed mitigation strategies feel underdeveloped
>
> We agree that our proposed mitigations are not definitive solutions. However, 1) we are one of the first to provide insights on the relationship between context use and vulnerability to prompt injection attack, based on which we introduce and show the effectiveness of the corresponding mitigation method built upon PINE, and 2) the primary contribution of our paper is not in proposing new methods but in systematically identifying and analyzing a significant security vulnerability in RAG systems.
> More fundamental architectural changes, such as sophisticated compartmentalization of information, are indeed promising directions that we highlight as future work. Our current focus is on providing actionable, minimally invasive defenses that can be immediately implemented in most existing RAG systems. While our mitigation strategies are indeed preliminary, their purpose is to serve as practical and accessible baselines and we view them as a starting point for further exploration, as is commonly seen in early-stage study on emerging security issues.
>
> > How do you envision fundamental prevention of this kind of attack in RAG systems?
>
> We think that fundamentally preventing such attacks might require rethinking the way RAG systems manage and compartmentalize retrieved information. For example, the backbone language model could be trained in a context-aware manner to discern retrieved content from user queries during processing, ensuring not following user instructions that target retrieved information. On the other hand, the RAG system could be designed to manage the user queries and the retrieved content in a more modularized manner such that the language model could tell apart information from different sources.
>
> > Have you considered more sophisticated architectural approaches to information compartmentalization?
>
> Yes, and we agree that these approaches represent a promising direction. For instance:
> 1) Multi-stream architectures: Employing separate attention streams for user queries and retrieved content might minimize unintended leakage.
> 2) Hierarchical processing: It involves structuring the flow of information within the model in distinct layers or stages, each dedicated to specific tasks, e.g. to interpret user queries or to retrieve/process/summarize datastore content.
> These preliminary ideas are part of our intended future research trajectory.
>
> **Final remarks**
>
> We appreciate your recognition of our empirical rigor and presentation clarity. While we acknowledge that our paper does not introduce fundamental theoretical contributions, we believe that our work makes a meaningful and timely impact by systematically studying an underexplored but critical vulnerability in RAG systems. Our findings serve as a reminder to the community of the trade-offs inherent in designing systems optimized for instruction-following while maintaining security.

---

> ### Comment · Reviewer_VHqF · 2024-11-27
>
> I have considered your comments and rebuttal, and I still believe that this paper is a relatively marginal contribution, which does deserve an acceptance but not necessarily a spotlight or oral.
>
> As such, I keep my main score (slightly raised the others), but I further urge the PC/SAC/AC to accept this paper if they feel "on the fence" about it. I'd give a 6.5 or 7 if I could.

---

### Official Review · Reviewer_zroi · 2024-11-03

**Soundness:** 3
**Presentation:** 3
**Contribution:** 4
**Rating:** 8
**Confidence:** 3

**Summary:**

This paper presents a study on prompt-injected data extraction from RAG systems, and showcases that both open-sourced and production RAG systems are vulnerable to data extraction attacks via simple approaches. Experiments and analysis showing the issue w.r.t, open-sourced LLMs and prod systems, as well as various system configurations. Attempts to mitigate the issues are also experimented and presented in the paper.

**Strengths:**

This paper studied prompt-injected data extraction from RAG system. This problem is of significant importance to practitioners and researchers, for awareness and developing mitigation strays. It’s also important for policymakers to recognize the risks lies in such systems. The paper analyzes the problems for open-source LLMs with different sizes, as well as production RAG systems. Additional experiments on various system configurations are also included.

**Weaknesses:**

The experiment set up might be overly simplistic. For open-sourced models, more sophistic RAG setup with prompts that intended to avoid data extraction (that goes beyond the simple system prompt 3.2.1) should be also experimented. Structured prompting format that clearly separate system prompts and user instructions can be more robust to such attacks. For production systems, GPTs are not necessarily representative of commercial production RAG systems, where guardrails preventing data leakages are put in various places of the pipeline. The paper will be significantly stronger if the approach shows effectives on more production RAG systems.

While the approach does clearly indicate data extraction from RAG systems/models is achievable, data extraction/leakage from a RAG system is not necessarily a concern. It will be helpful if the author can include a discussion section on various scenarios and clearly articulate for what scenarios there is a risk.

**Questions:**

L146: did you confirm with model provider that the Wikipedia articles are not included in the model training? Also, newer articles do not necessarily mean they are not overlapping with content in existing Wikipedia, so it’s a good idea to run a quick n-gram overlap check to remove articles that are not entirely new.

Section 3.2.2: Did you evaluate whether PINE affects RAG QA performance?

L429: what if we don’t have access to the GPTs’ system prompt (also leaked)?

---

> ### Author Response · Authors · 2024-11-20
> **Response to Reviewer zroi**
>
> We appreciate your thoughtful feedback and provide responses to the identified weaknesses and questions below:
> > The experiment setup might be overly simplistic
>
> We acknowledge the concern regarding the simplicity of the setup. In our study, we aimed to demonstrate the core vulnerability in a controlled manner to isolate and analyze the factors contributing to data extraction risks, thus deliberately choosing the simplest RAG design. We agree that exploring more sophisticated RAG setups, such as those involving structured prompts that separate system and user inputs more explicitly, would strengthen the paper and introduce more insights. Future iterations of this work will incorporate advanced prompting techniques and more robust setups as new mitigation strategies.
>
> > Scenarios where data extraction is a concern
>
> We appreciate the suggestion to expand on real-world scenarios. The risk of data extraction becomes particularly concerning in scenarios where RAG systems are integrated into applications dealing with sensitive, confidential, or proprietary data. For instance:
> - Enterprise Data Systems:
> RAG systems are often used in corporate environments where datastores include trade secrets, internal documentation, or confidential client information. An adversary extracting such data could lead to significant financial losses, breaches of confidentiality agreements, and reputational damage.
> - Healthcare and Finance Applications:
> In sectors like healthcare and finance, RAG systems may retrieve patient records, financial statements, or other regulated information. The unauthorized disclosure of such data could violate privacy laws (e.g., GDPR in Europe) and expose institutions to legal and financial penalties.
> - Consumer Applications:
> Personalized applications, such as virtual assistants or customer service bots, may access user-uploaded files or personal data. Extraction attacks here could expose users to identity theft or other forms of exploitation, especially if sensitive information (e.g., passwords, addresses) is stored in the datastore.
>
> > Did you confirm with model provider that the Wikipedia articles are not included in the model training? Also, newer articles do not necessarily mean they are not overlapping with content in existing Wikipedia, so it’s a good idea to run a quick n-gram overlap check to remove articles that are not entirely new.
>
> We selected Wikipedia articles created after November 1, 2023, to minimize overlap with the models’ training data. This cutoff date was chosen because most of the models tested in our study are unlikely to have been trained on data updated beyond this timeframe. This approach has also been validated by prior published work, such as Shi et al. [1], which similarly used recent and obsolete wikipedia data to construct benchmarks for membership inference attacks. Admittedly, this ensures minimal overlap but does not guarantee absolute exclusion of overlapping content, but we consider this to be a reasonable assumption and have minimal effects on obtaining meaningful experiment results.
>
> [1] Shi, W., Ajith, A., Xia, M., Huang, Y., Liu, D., Blevins, T., ... & Zettlemoyer, L. (2023). Detecting pretraining data from large language models. arXiv preprint arXiv:2310.16789.
>
> > Did you evaluate whether PINE affects RAG QA performance?
>
> No, because PINE work has already done such experiments in the “ Results on Retrieval-Augmented Question-Answering” section, and the results show that PINE is performing better than baselines.
>
> > what if we don’t have access to the GPTs’ system prompt (also leaked)?
>
> In our work, the GPTs are attacked in a two-stage manner, where in the first stage, we obtain the system prompt by simple prompt injection attack, and then in the second stage, we leverage the information in the system prompt to attack it.
> In future scenarios where the system prompt of GPTs is not directly accessible, the attack may require adjustments tailoring to the specific design of GPTs or other information. As our attack on GPTs is just a proof-of-concept to show how instruction following abilities could be leveraged to leak data, we did not dig further into such cases.

---

> > ### Comment · Reviewer_zroi · 2024-11-26
> >
> > I appreciate the thorough responses. Based on the discussion so far, I will keep my score.

---

### Official Review · Reviewer_K6HL · 2024-11-04

**Soundness:** 3
**Presentation:** 2
**Contribution:** 2
**Rating:** 5
**Confidence:** 5

**Summary:**

This paper investigates prompt-injected data extraction and reveals that instruction-tuned language models are vulnerable to data extraction via copying their contexts. With stronger instruction-following capability, the vulnerability increases. The paper also studies several mitigation strategies in response to the prompt-injected data extraction attacks.

**Strengths:**

1. The ablation study in Section 3.1 contains interesting findings (e.g., effect of chunking decisions and retrieved context size) that potentially benefit the privacy and security domains.

2. The paper analyzes attacking production LLMs such as ChatGPT, which might be of interest for many researchers.

**Weaknesses:**

1. The paper does not seem to introduce any novel mitigation strategies. One strategy is about following a simple prompt, and another strategy is about utilizing Position-Insensitive Encoding (PINE) from a previous work.

2. The paper does not investigate whether the choice of indexing and retrieval mechanism (e.g., "HippoRAG: Neurobiologically Inspired Long-Term Memory for Large Language Models" and "From Local to Global: A Graph RAG Approach to Query-Focused Summarization") would affect the attack results. Attacks on state-of-the-art RAG pipelines (not just using a good LLM) need to be conducted and analyzed. Otherwise, the impact of this paper is going to be limited.

3. Readability of some paragraphs (e.g., line 47 to 67 and line 452 to 476) is low due to their lengthy nature. I suggest breaking them into separate paragraphs or shortening some sentences.

**Questions:**

1. For the results and findings presented in the paper, were you performing attacks on naive RAG?

2. What will be the attack results when you apply the prompt-injected data extraction attacks on GraphRAG of "From Local to Global: A Graph RAG Approach to Query-Focused Summarization"?

---

> ### Author Response · Authors · 2024-11-20
> **Response to Reviewer K6HL**
>
> Thank you very much for your comments and advice. We appreciate the opportunity to clarify and strengthen our paper based on this feedback. Here we respond to each point:
>
> > The paper does not seem to introduce any novel mitigation strategies.
>
> We acknowledge your concerns about the two mitigation strategies. However, 1) we are one of the first to point out the relationship between context use and vulnerability to prompt injection attack, based on which we introduce and show the effectiveness of the corresponding mitigation method built upon PINE, and 2) the primary contribution of our paper is not in proposing new methods but in systematically identifying and analyzing a significant security vulnerability in RAG systems. Similar to other works in ML security [1,2,3], our study emphasizes the importance of a thorough exploration of a security issue and demonstrates baseline mitigations as a practical starting point. We consider a thorough analysis of a safety problem to be a complete study and provide foundational insights for future study, and advanced mitigation techniques could be an independent extension work.
>
> [1] Carlini, N., Tramer, F., Wallace, E., Jagielski, M., Herbert-Voss, A., Lee, K., ... & Raffel, C. (2021). Extracting training data from large language models. In 30th USENIX Security Symposium (USENIX Security 21) (pp. 2633-2650).
>
> [2] Carlini, N., Hayes, J., Nasr, M., Jagielski, M., Sehwag, V., Tramer, F., ... & Wallace, E. (2023). Extracting training data from diffusion models. In 32nd USENIX Security Symposium (USENIX Security 23) (pp. 5253-5270).
>
> [3] Nasr, M., Carlini, N., Hayase, J., Jagielski, M., Cooper, A. F., Ippolito, D., ... & Lee, K. (2023). Scalable extraction of training data from (production) language models. arXiv preprint arXiv:2311.17035.
>
> > The paper does not investigate whether the choice of indexing and retrieval mechanism would affect the attack results.
>
> Thank you for emphasizing the potential impact of indexing and retrieval mechanisms on attack outcomes. We fully agree that expanding our analysis to include alternative RAG architectures would enhance the scope and depth of our findings. This is an avenue we plan to explore in future work.
>
> In this submission, however, we intentionally focused on the simple but representative and reproducible RAG setups to prioritize scalability and reproducibility and to investigate how vulnerabilities evolve with model size. As noted in the introduction, we centered our study on a widely-adopted and established RAG method, the retrieve-in-context (RIC) manner, which is adopted as a canonical setting in various real-world RAG systems like LlamaIndex and LangChain. While many diverse architectures exist, they ultimately rely on LLMs as their core component and, as such, remain fundamentally vulnerable to prompt-injection attacks—a point we aim to illustrate in this work.
>
> > Readability of some paragraphs (e.g., line 47 to 67 and line 452 to 476) is low due to their lengthy nature. I suggest breaking them into separate paragraphs or shortening some sentences.
>
> We appreciate your feedback on readability. The sections you mentioned (lines 47–67 and 452–476) will be revised to improve clarity and conciseness. We will divide lengthy passages into shorter paragraphs and simplify complex sentences where appropriate.
>
> > For the results and findings presented in the paper, were you performing attacks on naive RAG?
>
> Yes, as stated in the introduction, our experiments were conducted on RAG systems using the widely-adopted RIC method, i.e. naive RAG. We deliberately chose this setup to highlight vulnerabilities in a straightforward and representative implementation. This choice helps ensure the reproducibility and applicability of our results across a broad range of RAG systems employing similar designs.
>
> > What will be the attack results when you apply the prompt-injected data extraction attacks on GraphRAG?
>
> We acknowledge the value of exploring attacks on GraphRAG and other SOTA RAG architectures. While our study emphasizes the vulnerability of naive RAG systems, the fundamental principles of prompt-injection attacks and their exacerbation with model scaling should extend to other configurations as long as there are LLMs engaged, including GraphRAG, but the attack might be less effective since GraphRAG includes a summarization stage that preprocesses the text sources and thus some information could not be copied verbatim. We plan to conduct additional experiments in this direction and hypothesize that systems integrating advanced graph-based indexing might still inherit similar vulnerabilities due to their reliance on LLMs.

---

> > ### Author Response · Authors · 2024-11-24
> > **Follow-up on our previous response**
> >
> > Thank you once again for your thoughtful review and valuable feedback! As we approach the end of the discussion period, we want to ensure that our previous responses have fully addressed all your concerns. If you have any additional questions or unresolved issues that we can clarify to achieve a better score, please don’t hesitate to let us know. We’re more than happy to assist further!

---

> > > ### Comment · Reviewer_K6HL · 2024-11-27
> > >
> > > Thanks a lot for your explanation! Based on your rebuttal, I cannot increase my scores and believe this paper is below the acceptance threshold.
> > >
> > > Overall, I think the contribution/impact of this paper is limited. You said this paper is about "systematically identifying and analyzing a significant security vulnerability in RAG systems". Then it is fine not trying novel mitigation techniques, but readers would like to see a variety of widely-adopted and established RAG methods to understand the generalizability of your paper. I mentioned GraphRAG, because this method seems to be a popular and established example (along with several other indexing and retrieval methods). Especially when you said that attacks on GraphRAG might be less effective, I have concerns over the impact of your analysis. I guess people building real applications would use a somewhat more advanced tool than naive RAG. You cannot simply say that you will leave this work for future analysis and still claim a systematic study.
> > >
> > > Please note that I am not asking for new experiments, because this comment is the same as what I wrote at first.

---

> > > > ### Author Response · Authors · 2024-11-27
> > > > **Response to Reviewer K6HL**
> > > >
> > > > Thank you for your additional feedback!
> > > >
> > > > We recognize that the inclusion of popular architectures such as GraphRAG would strengthen our argument for the ubiquity of the vulnerabilities. However, we maintain that our focus on Retrieve-In-Context (RIC)-based systems in this work is a deliberate choice. This setup provides a clear and reproducible baseline for studying scaling effects on data leakage vulnerabilities and aligns with widely-deployed real-world systems as mentioned in our paper.
> > > >
> > > > As for “people building real applications would use a somewhat more advanced tool than naive RAG”: The naive RAG method remains a default setting of RAG setup and is still used in many real-world commercial use [1,2,3]. Besides, about popular RAG papers, take GraphRAG for example, it essentially changes how knowledge is indexed and queried but does not change the interface between retrieved information and the LLM, which is exactly the component that a naive RAG setup exposes. In our updated draft we will refer to and discuss attacks on GraphRAG, but we reiterate that: While many diverse RAG architectures exist, they ultimately rely on prompting LLMs as their core component and, as such, remain fundamentally vulnerable to prompt-injection attacks—a point we aim to illustrate in this work.
> > > >
> > > > With regard to "not trying novel mitigation techniques": we consider the discovery of relationships between position bias and prompt-injected data extraction attacks to be novel. Although we did not introduce the position bias elimination method itself, our application of it as a mitigation strategy is new and has not been proposed elsewhere.
> > > > In conclusion, we appreciate your input and will incorporate these clarifications into our updated draft to better communicate the scope and contributions of our work.
> > > >
> > > > Thank you again for your feedback!
> > > >
> > > >
> > > > [1] VoyageAI. Voyageai. 2024. URL https://www.voyageai.com/.
> > > >
> > > > [2] LangChain. Langchain, 2022. URL https://www.langchain.com/.
> > > >
> > > > [3] LlamaIndex. Llamaindex, 2024. URL https://www.llamaindex.ai/.

---

> > > > > ### Comment · Reviewer_K6HL · 2024-12-02
> > > > >
> > > > > Thanks again for your response! I carefully checked your draft and the comments from other reviewers again, and I have increased my confidence to 5. The paper definitely has merit, but I still think its impact/contribution is limited.
> > > > >
> > > > > When people utilize a different RAG architecture (built by additional LLM calls), they are not directly prompting LLMs using their corpus anymore. Thus, unlike naive RAG, I assume prompt-injection attacks would be less effective, which is also acknowledged by the authors. These RAG architecture preprocesses people's corpus in a way that might improve overall security level. I think it is too assertive to justify that the paper is a "systematic study".
> > > > >
> > > > > Moreover, the authors emphasize that they are pursuing a clear and reproducible baseline, but the tools I suggested are also clear. I am not sure if those tools are 100% reproducible, but we can fix their knowledge graph since we only need to run indexing once for our data. In that case, those tools are reproducible. The authors claim that they intentionally chose naive RAG, which I think needs further exploration.

---

### Meta-Review · Area_Chair_HKqR · 2024-12-21

**Metareview:**

The paper investigates adversarial attacks to extract the retrieval corpus of a RAG pipeline. It shows that simple prompt injection attacks can extract 41% of a 77K words book with only 100 generated queries. Overall, the reviewers find the analysis in the paper although the attack mitigation strategies proposed are quite limited. A more extensive exploration of both attacks and mitigation strategies would strengthen the paper. Extending the analysis to RAG systems beyond the simple RIC framework would also be useful.

**Additional Comments On Reviewer Discussion:**

Most of the reviewer comments are about extending the scope of the study, either to more sophisticated RAG systems, attack and mitigation strategies. Overall, the reviewers would like more analysis, but accept that the paper in its current state makes for a valuable contribution.

---

### Decision · Program_Chairs · 2025-01-22

Accept (Poster)